# Concept of Fluorescent Transport Activity Biosensor for the Characterization of the Arabidopsis NPF1.3 Activity of Nitrate

**DOI:** 10.3390/s22031198

**Published:** 2022-02-04

**Authors:** Yen-Ning Chen, Cheng-Hsun Ho

**Affiliations:** Agricultural Biotechnology Research Center, Academia Sinica, Taipei 115, Taiwan; burhdgfhan@trunojoyo.ac.id

**Keywords:** *Arabidopsis thaliana*, biosensor, fluorescent, transporter

## Abstract

The NRT1/PTR FAMILY (NPF) in Arabidopsis (*Arabidopsis thaliana*) plays a major role as a nitrate transporter. The first nitrate transporter activity biosensor NiTrac1 converted the dual-affinity nitrate transceptor NPF6.3 into fluorescence activity sensors. To test whether this approach is transferable to other members of this family, screening for genetically encoded fluorescence transport activity sensor was performed with the member of the NPF family in Arabidopsis. In this study, NPF1.3, an uncharacterized member of NPF in Arabidopsis, was converted into a transporter activity biosensor NiTrac-NPF1.3 that responds specifically to nitrate. The emission ratio change of NiTrac-NPF1.3 triggered by the addition of nitrate reveals the important function of NPF1.3 in nitrate transport in Arabidopsis. A functional analysis of Xenopus laevis oocytes confirmed that NPF1.3 plays a role as a nitrate transporter. This new technology is applicable in plant and medical research.

## 1. Introduction

Nitrogen is one of the most limiting nutrients for plants. The maximal yield of a plant depends on nitrogen acquirement and distribution. Improvements in the nitrogen use efficiency of plants are urgently required, considering the increasing human population and the threat of global warming. Nitrate serves as one of the major forms of nitrogen for plants. Nitrate is first taken up from the soil into the roots [1]. Once nitrate is in the cells, it can either assimilate to nitrite by nitrate reductase and subsequently be reduced to ammonium by nitrite reductase and amino acids or be stored in vacuoles to help maintain the osmotic balance [2,3]. Besides serving as a nutrient source, nitrate also functions as a signal molecule regulating plant metabolism and development [4,5,6]. Nitrate plays a key role in the regulation of plant growth and development; the regulation of nitrate uptake and nitrate transport helps plants to mediate their growth in response to differing environmental conditions.

There are many factors that affect nitrate uptake, such as the nitrogen level, the energy status of the plant, the assimilation status of imported nitrogen, the N-demand, and the mobile signals between shoots and roots as well as between different parts of the root system. Thus, it is important to understand or even monitor the activity of transporters in the cells of plant roots in a minimally invasive manner. Genetically encoded fluorescent nanosensors have proven to be valuable for monitoring ions and metabolite levels with a high spatial and temporal resolution in a minimally invasive way [7]. These sensors rely on conformational rearrangements in a sensory domain triggered by substrate binding. These rearrangements are further affected by changes in the Förster Resonance Energy Transfer (FRET) efficiency between two fluorescent proteins, which act as FRET donors and acceptors due to their spectral overlap. Many sensors, such as glucose, have been used in Arabidopsis to monitor steady-state levels, accumulation, and elimination under both static and dynamic conditions where roots have been exposed to different analytes [8,9,10,11,12]. 

Such rearrangements have recently been observed for the members of the NPF/POT family [13]. AmTrac is the first prototype for transport activity sensors, which use ammonium transporters as sensory domains for engineering transport activity sensors by inserting a circularly permutated EGFP (cpEGFP) into a conformation-sensitive position of an ammonium transporter [8]. The addition of ammonium to yeast cells expressing the AmTrac sensor triggers concentration-dependent and reversible changes in fluorescence intensity [8]. The first transport activity sensors of NiTrac (nitrate transport activity) and PepTrac (peptide transport activity) for nitrate and peptides were created using CHL1/NRT1;1/NPF6.3 and PTRs for the members of the NPF family [14]. More recently, AmTryoshka1;3, a ratiometric Matryoshka biosensor from a nested cassette of green- and orange-emitting fluorescent proteins, was further developed and introduced into plants [15].

Nitrate uptake is complex, and the major families of transporters used for nitrate uptake and distribution, low-affinity/high-capacity and high-affinity/low-capacity systems, have been identified [16,17,18]. The NPF/POT nitrate transporter family [19], NRT2 nitrate transporters [20], are known to mainly be involved in nitrate uptake from the soil solution into root cells. In *Arabidopsis*, there are 53 NPF transporters. However, less than twenty of the members of this family have been functionally characterized in the past few decades. For example, NRT1.5 mediates nitrate efflux and is expressed in root pericycle cells [21]. *NRT1.8* is expressed in root xylem parenchyma cells and is involved in transport nitrate from the xylem sap [22]. NRT1.7 mediates the remobilization of nitrate from older leaves to younger leaves [23]. NRT1.9 mediates the phloem loading of nitrate [24]. NRT1.13 mediates shoot architecture and flowering [25], NRT1.11 and NRT1.12 mediate the redistribution of nitrate in xylem cells [26], and NRT1.4 affects the development of leaves [27], etc. In addition to their role in nitrate uptake, members of the NPF/POT [19] family also play important roles in the transport of histidine; dicarboxylates; oligopeptides; glucosinolates; and phytohormones—i.e., auxin, ABA, and gibberellin [28,29,30]. Thus the functional characterizations of other unknown members of the NPF family is important; additionally, whether the use of a transport activity sensor is transferable to other proteins of this family remained to be shown. To create transport activity sensors, a screen that fused members of the NPF family to fluorescent protein pairs, expressed the fusions in yeast, and tested their response to nitrate addition was performed. The NiTrac-NPF1.3 biosensor responded to the addition of nitrate. The response of NiTrac-NPF1.3 sensor to nitrate was specific and reversible. Furthermore, the functional analysis of NPF1.3 expressing *Xenopus laevis* oocytes confirmed that NPF1.3 plays a role as a nitrate transporter. The successful use of this sensor in yeast indicates that this new tool can be used for in planta analyses.

## 2. Materials and Methods

### 2.1. DNA Constructs

The DNA constructs were analyzed as described previously [14]. Vector constructions have been described previously [31]. In brief, all constructs were inserted using Gateway LR reactions into the yeast expression vector pDRFlips-GW. pDRFlip39 contains an N-terminal enhanced dimer Aphrodite t9 (edAFPt9) and C-terminal fluorescent protein enhanced dimer, with 7 amino acids and 9 amino acids truncated from the N-terminus and C-terminus of eCyan (t7.ed.eCFPt9), respectively. The full length of NPF1.3 (At5g11570) as well as the ORFs of the NPF from Arabidopsis were used as sensory domains for creating the sensor. For functional assays in *Xenopus* oocytes, the cDNA of NPF1.3 was cloned into the oocyte expression vector pOO2-GW [32].

### 2.2. Yeast Cultures

The yeast culture was performed as described in [33]. The yeast strains BJ5465 [MATa, ura3–52, trp1, leu2Δ1, his3Δ200, pep4::HIS3, prb1Δ1.6R, can1, and GAL+] were used for further sensor response assays. Yeast was transformed using the lithium acetate method [34] and transformants were further cultured on solid or liquid YNB (Difco) supplemented with 2% glucose and –*ura* DropOut medium (Clontech).

### 2.3. Fluorimetry

The fluorimetry assays were performed as described previously [14]. In brief, yeast cultures were washed three times in 50 mM of MES buffer at pH 5.5 and resuspended in the same MES buffer. The fluorescence of the sensors was measured using a fluorescence plate reader (M1000, TECAN, Grödig, Austria) in bottom reading mode using a 7.5 nm bandwidth for both excitation and emission [35,36]. The emission spectra were recorded (λ_em_ 470–570 nm) in 96-well flat-bottom plates (#655101; Greiner, Monroe, NC, USA). Fluorescence from cultures harboring 39 donor: t7.ed.eCFPt9 was measured by excitation at λ_exc_ 428 nm.

The quantitative fluorescence intensity from individual yeast cells was determined as described previously [14]. Quantitative fluorescence intensity data were acquired on an inverted microscope (Leiss, LSM, 510). Yeast cells were cultured in a microfluidic perfusion system (Y04C plate, Onyx, Cellasic, Hayward, CA, USA) and perfused with either 50 mM of MES buffer at pH 5.5 or buffer supplemented with 10 mM of KNO_3_ [35,37]. Briefly, imaging was performed on an inverted fluorescence microscope (Leiss, LSM, 510) with a QuantEM digital camera (Photometrics) and a 40×/NA (numerical aperture) 1.25–0.75 oil-immersion lens (IMM HCX PL Apo CS). Then, 440 nm and 514 nm lasers were used to excite cyan fluorescent protein (CFP) and YFP, respectively. Fluorescence emissions were detected by a PMT detector at 463 to 508 nm for CFP and at 520 to 585 nm for YFP. The power of the lasers was set between 0.5% and 2% to image CFP or YFP. The Fiji software as well as ROI manager tool were used to quantify the fluorescence intensity.

### 2.4. Functional Expression of NPF1.3 in Xenopus Oocytes

TEVC in oocytes was performed as described previously [8]. For in vitro transcription, pOO2-NPF1.3 was linearized with *Mlu*I. cRNA was transcribed by SP6 RNA polymerase using the mMESSAGE mMACHINE kits (Ambion, Austin, TX, USA). The oocytes were injected via the Roboocyte (Multi Channel Systems, Reutlingen, Germany; [38,39]) with distilled water (50 nL as control) or cRNA from NPF1.3 (50 ng in 50 nL). Oocytes were kept at 16 °C for two to four days after being injected with ND96 buffer (96 mM NaCl, 2 mM KCl, 1.8 mM CaCl_2_, 1 mM MgCl_2_, and 5 mM HEPES, pH 7.4, gentamycin (50 μg/μL)) before beginning the experiments.

### 2.5. Electrophysiological Measurements in Xenopus Oocytes

Electrophysiological analyses of injected oocytes were performed as described previously [8,40]. Reaction buffers were: (i) 230 mM mannitol, 0.3 mM CaCl_2_, and 10 mM HEPES; (ii) 220 mM mannitol, 0.3 mM CaCl_2_, and 10 mM HEPES at the pH indicated plus 5 mM CsNO_3_. A two-electrode voltage-clamp (TEVC) Roboocyte system (Multi Channel Systems) was used for current recordings [38,39].

### 2.6. Statistical Analyses

We used the analysis of variance (ANOVA) for all statistical analyses; factors (sample, treatment) were treated as fixed factors. ANOVAs were performed using the Analysis of Variance (ANOVA) Calculator for One-Way ANOVA from Summary Data (www.danielsoper.com/statcalc, accessed on 21 December 2021). The reported values represent means and standard deviations. Student’s *t*-test was used to determine significance. All experiments were performed with at least three biological repeats.

## 3. Results

### 3.1. A Screen of Engineering of Members of NPF into Transport Activity Sensor

Engineering FRET biosensors often requires the screening of ligand-binding domains fused to pairs of FRET donor/acceptor variants. To capture the substrate-dependent conformational rearrangements of transporters, NiTrac1 and PepTracs were generated by sandwiching CHL1/NRT1;1/NPF6.3 and PTRs between a yellow acceptor and cyan donor fluorophore [14]. Based on our previous successful experiments with NiTrac1 and PepTracs [14], it is likely that the other members of the NPF family undergo conformational rearrangements during their transport cycle. To measure conformational rearrangements during the transport cycle, screening in which members of NPF were sandwiched between enhanced dimerization (ed) variants of Aphrodite (edAFP) as FRET acceptors and edeCFP as FRET donors capable of FRET was performed [41]. The workflow diagram used for engineering a biosensor and testing is shown in Figure 1.

These chimeras were expressed in yeast, followed by the spectral analysis of yeast cultures in a spectrofluorimeter. The fluorophores of a chimera, named NiTrac-NPF1.3 (Figure 2a), were in the Förster distance, and the emission ratio changes were induced by nitrate addition, as we expected a change in the energy transfer rate between the emission at 530 nm (donor excitation acceptor emission, DxAm) and a concomitant change in the emission at 488 nm (donor excitation donor emission, DxDm) as a FRET ratio change sensor (ΔDxAm/DxDm) (Figure 2b). On the basis of NFP1.3 belonging to the NFP family, one hypothesis is that NiTrac-NPF1.3 shows a FRET ratio change upon binding to nitrate.

### 3.2. Selectivity of NiTrac-NPF1.3 Biosensor

The NPF/POT family are characterized as nitrate or peptide transporters [19]. More recently, other ions or hormones have also been reported to be transported by NFPs [28,29,30]. To test whether the emission ratio changes are specific to nitrate but not triggered by other ions or peptide, compounds such as ammonium, MgCl_2_, CaCl_2_, and dipeptide and different concentrations of nitrate were further tested in NiTrac-NPF1.3-expressing yeast cells. The response of NiTrac-NPF1.3 is nitrate-specific; other compounds such as ammonium, divalent cations, and dipeptide showed no significant effect (Figure 3). Together, these results strongly support the hypothesis that NiTrac-NPF1.3 measures nitrate concentrations and/or reports conformational state changes of the transporter, suggesting that NPF1.3 may function as a nitrate transporter in Arabidopsis.

### 3.3. A Single Cell Measurement of NiTrac-NPF1.3 Biosensor

To study the dynamic response and mechanism of NiTrac-NPF1.3 in more detail, the emission ratio response of NiTrac-NPF1.3 to nitrate was tested in single yeast cells [14]. The emission ratio of NiTrac-NPF1.3 in individual yeast cells changed fast with the addition of external nitrate and back to the basal level when external nitrate was reduced by washing it away (Figure 4). The measurement of the sensor response in individual yeast cells demonstrated a rapid nitrate-induced emission ratio change and the reversibility of the fluorescence intensity after the removal of nitrate, indicating that the sensor can be used effectively for in planta analyses.

### 3.4. NPF1.3 Functions as Nitrate Transporter

The above experiments showed that the emission ratio response of NiTrac-NPF1.3 was related to the nitrate added, suggesting that the putative membrane protein NPF1.3 could function as a nitrate transporter. To elucidate the function of NPF1.3, NPF1.3 was heterologously expressed in *Xenopus laevis* oocytes and two-electrode voltage clamping assay was used for the substrate specificity analysis. As shown in Figure 5, the nitrate-dependent inward currents were observed by KNO_3_ in *NPF1.3*-injected oocytes when compared with water-injected oocytes, but no current was elicited by KCl. These data suggest that NPF1.3 plays a role as a nitrate transporter in plasma membrane. Together, these data show that NiTrac-NPF1.3 can be used as a tool for sensing and reporting conformational changes in nitrate transport and further support the hypothesis that NiTrac-NPF1.3 reports the activity states of the transporter.

## 4. Discussion

To be able to characterize and monitor the activity and regulation of individual isoforms of the NPF transporter family in Arabidopsis, screening of a genetically encoded fluorescence transport activity sensor was performed We further characterized and demonstrated in vitro that NPF1.3 is a nitrate transporter. In addition, we engineered NPF1.3 to be a transport activity biosensor to report its activity and conformation in vivo. The NPF1.3 protein of the NPF family was fused with yellow and cyan versions of GFP at their N- and C-termini, respectively. When expressed in yeast, the NiTrac-NPF1.3 sensor responded to substrate addition with a FRET change. The response of NiTrac-NPF1.3 to nitrate is specific and reversible. The data of the two-electrode voltage clamping assay in *Xenopus laevis* oocytes provide further evidence of the role of NPF1.3 as a nitrate transporter in Arabidopsis. Together with our studies, the engineering of activity sensors through the fusion of fluorophores to the NPF/POT family proteins indicates the potential to transfer this concept to other transporters, receptors, and enzymes. This suite of genetically encoded sensors provides a unique set of tools for observing the activity of individual transporter family members in intact tissue layers of intact plants.

In addition to the cytosolic sensor, the transport activity sensors can further report information on structure, such as the basic ratio of the sensor. The specific conformation of the population of sensors can be compared between, for example, mutants or in response to the coexpression of a regulator. This idea is supported by previous studies of the transport activity of sensors by introducing point mutations into NiTrac1 as well as co-expressing putative interactors [14]. For example, by introducing mutations into NiTrac1, the residues which are responsible for uptake and different affinities to nitrate can be identified—e.g., E41A, E44A, R45A, and E476A of NPF6.3 are related to uptake activity, while T101A and R264A/R266A/K267A are related to the affinity of nitrate. The NiTrac1 sensor response to nitrate is affected by the co-expression of CIPK23 and CBL1 in yeast, indicating that CIPK23 and CBL1 play a role as regulators of NPF6.3 [14]. Taking advantage of a homology model, it will be interesting to further explore the putative interactors as well as those residues conserved in the binding pocket and/or salt bridge, which is important for the substrate transport cycle in NiTrac-NPF1.3. Without the data from the mutants of the conformational changes, we nevertheless provide evidence that the activity sensor is highly sensitive and could be used as a simple tools for probing structure–function relationships in heterologous and homologous systems without the necessity of purifying the transporters.

In the past few decades, many members of the NPF family have been identified and characterized in Arabidopsis. Some of these are primary root expression dominant, while some are highly expressed in shoot or seeds. For example, the plasma membrane-localized low-affinity nitrate transporter *NPF7.3/NRT1.5* is expressed in root pericycle cells nearby the xylem and functions in nitrate loading in the xylem [21]. *NPF2.9*/*NRT1.9*, which is expressed in root companion cells, participates in the phloem loading of nitrate [24]. NPF5.11, NPF5.12, and NPF5.16, which are localized in tonoplasts, mediate nitrate efflux from vacuoles [42]. *NPF6.2/NRT1.4* is highly expressed in the petiole and midrib of leaves [27]. *NPF7.2/NRT1.8* is expressed in xylem parenchyma cells and mediates nitrate retrieval from the xylem [22]. NPF6.3/NRT1,1/CHL1 and NPF4.6/NRT1.2 are both highly expressed in the roots and responsible for dual-affinity and low-affinity nitrate uptake, respectively [19]. *NPF2.12*/*NRT1.6*, which is expressed in the vascular tissue of the silique, delivers nitrate to the embryo [43]. NPF2.13/NRT1.7, detected in the phloem of the leaf, remobilizes nitrate from old leaves to young leaves [23]. Besides the NRT1 proteins mentioned above, NRT2 proteins also participate in low nitrate uptake. *NPF2.4* is expressed mainly in the root stele and *NPF2.5* is expressed in the root cortical cells [44,45]. NRT2 proteins are exclusively responsible for nitrate uptake, while NPF proteins are involved in transport processes such as nitrate distribution between different tissues [19]. Despite many nitrate transporters having been identified, the nitrate transporter, which is responsible for the connection of nitrate transport between the epidermis and vascular tissues, is still unclear. In the *Arabidopsis* eFP browser (efp.ucr.edu) [46], it can be seen that NPF1.3 is dominantly expressed in the endodermis cells of the root in Arabidopsis. The next step will be to deploy NiTrac-NPF1.3 in Arabidopsis plants to characterize the activity of the transporter and its regulation in vivo. It will be interesting to further explore the interaction of NiTrac1 and NiTrac-NPF1.3 regarding the nitrate uptake and distribution in plants.

In addition to transporting nitrate, many studies suggest that the substrates of NPF transporters may vary, such as NPF8.1/PTR1, NPF8.2/PTR5, and NPF8.3/PTR2, which function as peptide transporters [47,48,49]. NPF2.4 mediates chloride loading into xylem, NPF2.5 mediates chloride efflux from the root [44,45], and NPF7.3 may function in the uptake of potassium [50]. Increasing numbers of reports show that NPFs can transport plant hormones or other secondary metabolites, such as transporting auxin, abscisic acid, gibberellins, and glucosinolates, by NPF6.3/NRT1.1, NPF4.1/AIT3, NPF3, and NPF2.10/NPF2.11, respectively [28,29,51,52]. It will be interesting to determine whether NPF1.3 could also transport hormones or other secondary metabolites in the future.

## 5. Conclusions

To sum up, by the extension the idea of an engineer transporter activity biosensor such as NiTrac1 [14] to the members of the NPF family, we developed the NiTrac1.3 biosensor to report the activity of nitrate transporter NPF1.3 in vivo. In the next steps of our research, we will deploy both NiTrac1 and NiTrac1.3 sensors in Arabidopsis roots to measure the nitrate fluxes in vivo, to study their regulation, and identify their regulators in vivo. The development of this activity biosensor and in vivo flux analyses will help us to understand the action of transporters and may also provide tools that can be used for drug development and discovery.

## Figures and Tables

**Figure 1 sensors-22-01198-f001:**
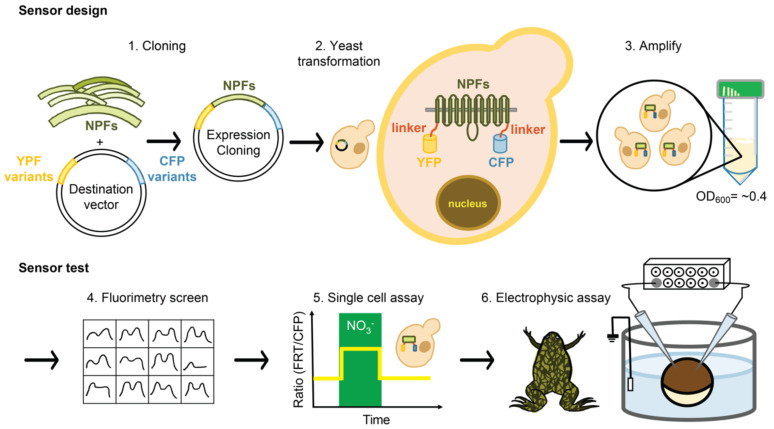
A workflow diagram of sensor development and measurement. The diagram, from step 1 to step 6, shows cloning strategies based on NiTrac1 [14] and screening in yeast cells and the testing of the sensor’s response to the substrate. NPFs were cloned into the vector with CFP and YFP proteins. After transformation and amplification with yeast, the sensor responses to nitrate were tested via a fluorimetry screen or individual single-cell assay in yeast or in the oocyte.

**Figure 2 sensors-22-01198-f002:**
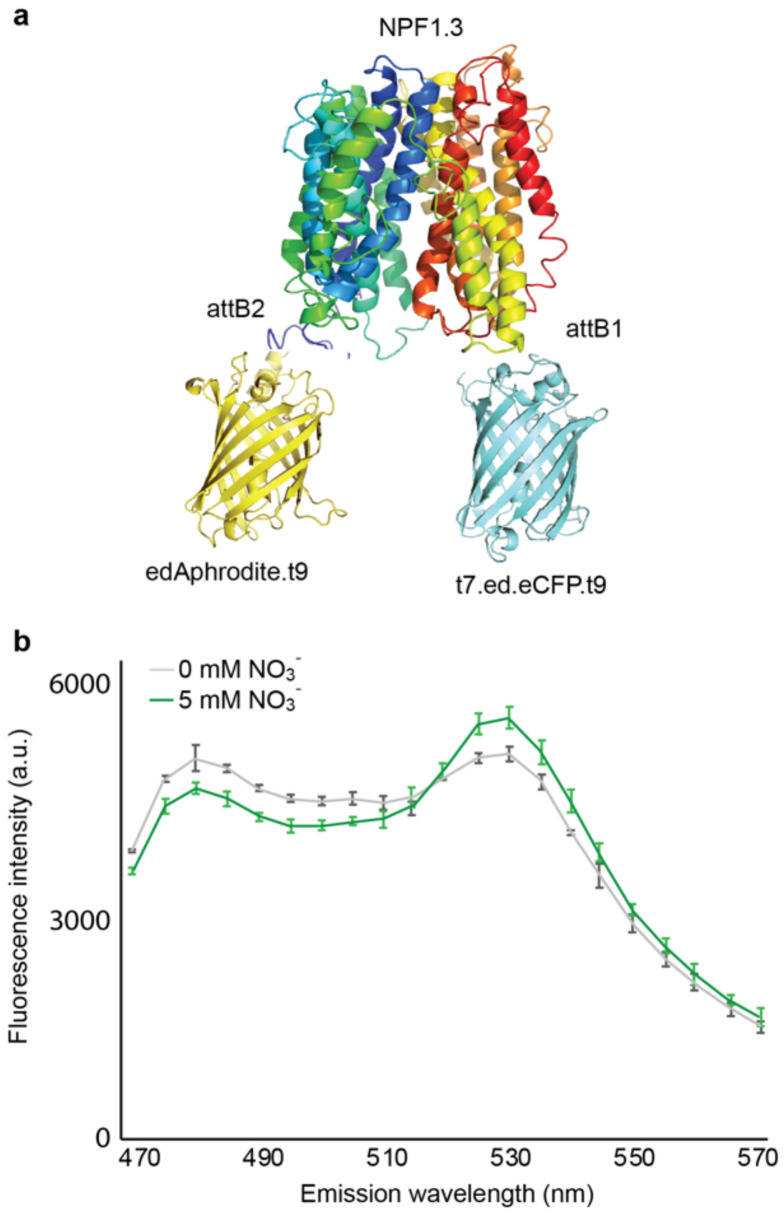
Design and development of the NiTrac-NPF1.3 sensor (**a**). Schematic representation of the NPF1.3 sensor construct. edAphrodite.t9, yellow; t7.ed.eCFP.t9, light blue; NPFs, colors. (**b**). Emission spectra for NiTrac-NPF1.3 expressed in yeast cells; excitation at 428 nm: the addition of 5 mM potassium nitrate (green; control buffer, gray) leads to an emission ratio change triggered by nitrate. Means and s.d. of three biological repeats are presented. Similar results were obtained in at least three independent experiments.

**Figure 3 sensors-22-01198-f003:**
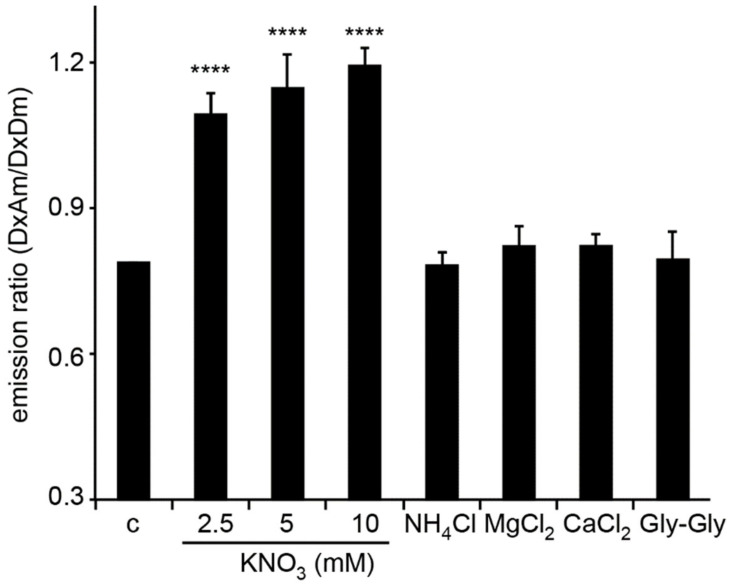
Emission ratio change of NiTrac-NPF1.3 specific to nitrate. Substrate specificity: Yeast cells expressing NiTrac-NPF1.3 were treated with the indicated compounds at 5 mM concentrations or as indicated in the figure. Only nitrate triggered responses that were significantly different from the control c (****, *p* < 0.0001, *t*-test). Experiment performed as in Figure 1. Nitrate triggers emission ratio changes with different nitrate concentrations. Nitrate-induced ratio change (peak fluorescence intensity of Aphrodite excited at 505 nm over emission spectrum at 485 nm obtained with excitation at 428 nm). Means and s.d.s of six biological repeats are presented (one-way ANOVA followed by Tukey’s post-test).

**Figure 4 sensors-22-01198-f004:**
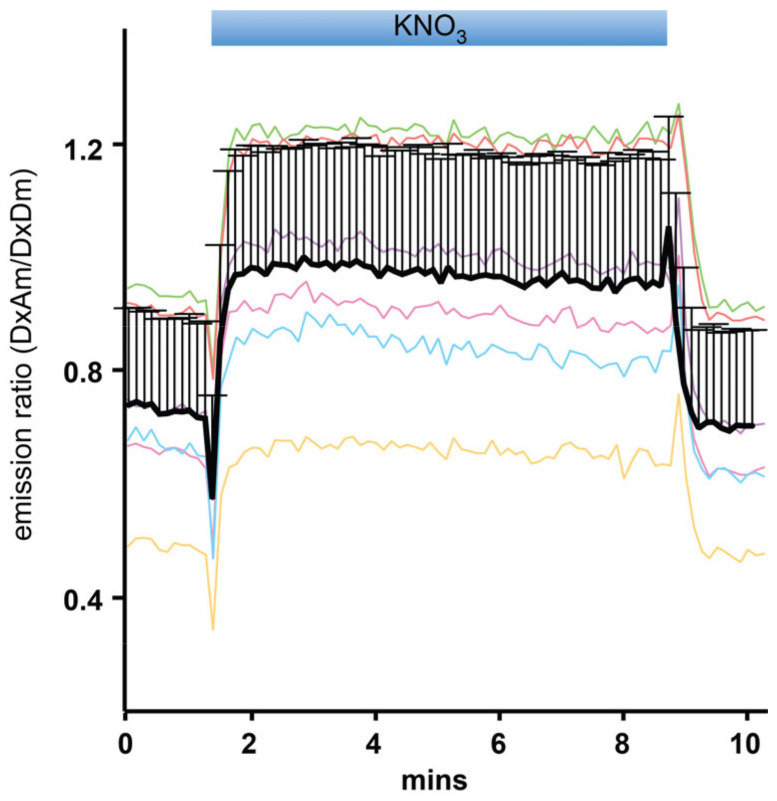
Analysis of the NiTrac-NPF1.3 response in individual yeast cells trapped in a Cellasic microfluidic plate. Cells were initially perfused with 50 mM of MES buffer at pH 5.5, followed by a square pulse of 5 mM of KNO_3_ in MES buffer for about seven minutes (blue frame). Color lines: individual data of cells. Black line: normalized data from individual cells (mean ± SD; *n* = 6). Similar results were obtained in at least three independent experiments.

**Figure 5 sensors-22-01198-f005:**
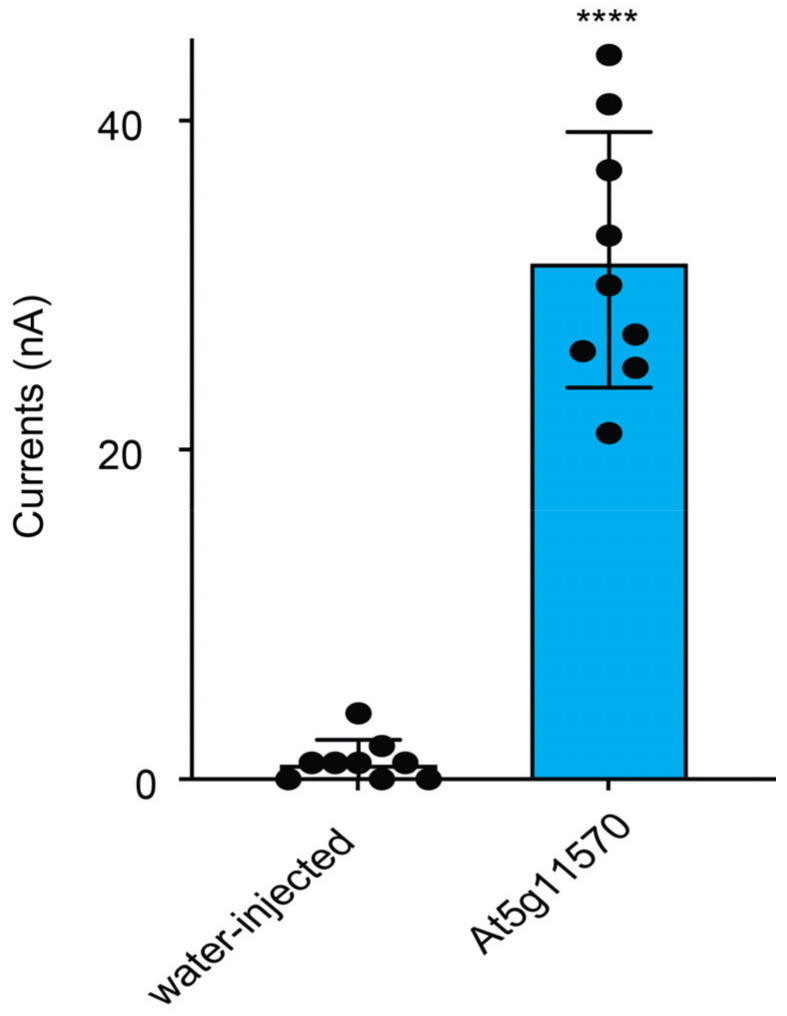
Currents of NPF1.3 using TEVC. Oocytes were injected with water only, cRNAs of NPF1.3, and perfused with CsNO_3_ at 5 mM for current recordings. Oocytes were voltage clamped at −120 mV. The data are the mean ± SE for three experiments. **** indicates the significant difference, *p* < 0.0001, *t*-test). Similar results were obtained in at least three independent experiments using different batches of oocytes (one-way ANOVA followed by Tukey’s post-test).

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
