# Peer review of "Concept of Fluorescent Transport Activity Biosensor for the Characterization of the Arabidopsis NPF1.3 Activity of Nitrate"

_sensors, 2022, doi:10.3390/s22031198_

Round 1

Reviewer 1 Report

The authors construct a nitrate transporter activity biosensor NiTrac1, which has been converted the dual-affinity nitrate transceptor NPF6.3 into fluorescence activity sensors. I think the results can't strong support this view. Some changes are not obvious. This is a transporter activity biosensor, how to nitrate transport? The present experimental results are not not sufficient. Therefore, I think this paper can not accepted in this journal.

Author Response

Comments and Suggestions from reviewer1:

The authors construct a nitrate transporter activity biosensor NiTrac1, which has been converted the dual-affinity nitrate transceptor NPF6.3 into fluorescence activity sensors. I think the results can't strong support this view. Some changes are not obvious. This is a transporter activity biosensor, how to nitrate transport? The present experimental results are not not sufficient. Therefore, I think this paper can not accepted in this journal.

Response to reviewer1:

Recently, the concept of transport activity sensor by using fluorescent proteins to monitor how the shape of the transporter change during its transport cycles has been evidently demonstrated and reported in vitro as well as in vivo including in planta, i.e., biosensor of AmTrac, MepTrac, NiTrac, PepTrac, as well as AmTryoshka (Ast et al., 2015; Ast et al., 2017; De Michele, R et al.,2013; Ho and Frommer, 2014). By taking advantage of the concept and to extend it to other membrane proteins, we here have successfully converted an unknown protein (a member of NPF family in Arabidopsis) into a sensor, NiTrac-NPF1.3, which specifically responses to the additon of nitrate in yeast. Like other nitrate transporters in NPF family, the nitrate-triggered inward currents has been also seen from the results of our TEVC assays in NPF1.3 expressed Xenpous oocytes indicating that NPF1.3 is a nitrate transporter. Together, these data provide additional evidence and support the concept of monitoring the shape of the transporter changes by using fluorescent proteins as well as NiTrac-NPF1.3 as a transport activity sensor for nitrate.

References:

Ast, C., De Michele, R., Kumke, M.U., and Frommer, W.B. (2015). Single-fluorophore membrane transport activity sensors with dual-emission read-out. eLife 4. ARTN e07113

10.7554/eLife.07113.

Ast, C., Foret, J., Oltrogge, L.M., De Michele, R., Kleist, T.J., Ho, C.H., and Frommer, W.B. (2017). Ratiometric Matryoshka biosensors from a nested cassette of green- and orange-emitting fluorescent proteins. Nat. Commun. 8, 431. 10.1038/s41467-017-00400-2.

De Michele, R., Ast, C., Loque, D., Ho, C.H., Andrade, S., Lanquar, V., Grossmann, G., Gehne, S., Kumke, M.U., and Frommer, W.B. (2013). Fluorescent sensors reporting the activity of ammonium transceptors in live cells. eLife 2, e00800. 10.7554/eLife.00800.

Ho, C.H., and Frommer, W.B. (2014). Fluorescent sensors for activity and regulation of the nitrate transceptor CHL1/NRT1.1 and oligopeptide transporters. eLife 3, e01917. 10.7554/eLife.01917.

Reviewer 2 Report

In this interesting paper, screening for genetically encoded fluorescence 10
transport activity sensor has been performed with the member of NPF family in Arabidopsis. Paper described the applicability of previously published approach of nitrate transporter activity biosensor NiTrac1,  to other members of this family. Paper is clearly presented and concluded. I have only one remarks. What was the motivation of selectivity tests towards as ammonium, MgCl2, CaCl2, and dipeptide? Are these compounds the only interferents?

Author Response

Response to Reviewer2:

Reviewer 2

In this interesting paper, screening for genetically encoded fluorescence 10
transport activity sensor has been performed with the member of NPF family in Arabidopsis. Paper described the applicability of previously published approach of nitrate transporter activity biosensor NiTrac1, to other members of this family. Paper is clearly presented and concluded. I have only one remarks. What was the motivation of selectivity tests towards as ammonium, MgCl2, CaCl2, and dipeptide? Are these compounds the only interferents?

The reason is to test the substrate specificity of the NPF1;3 as well as to test whether the response of NiTrac1;3 is triggered by dipeptide or other ions, regarding the members in NPF family have been reported transport other ions. Thus, several ions and dipeptide were chosen and be tested. To make it clearer, the sentence as “More recently, other ions or hormones have been reported can be transported by NFPs [29-31]. To test whether the emission ratio changes are specific to nitrate but not triggered by other ions or peptide…” in line 175 was added.

Reviewer 3 Report

Comments to the authors:

This paper reported the fabrication of fluorescent transport activity biosensor for characterization of Arabidopsis NPF1.3 activity of nitrate. Measurements of biosensor selectivity and additional experiments of nitrate transportation were performed. Overall, the work was well conducted, and the results were adequately presented and discussed. The manuscript can be considered for publication after minor corrections that should be revised by the authors.

Specific comments:

  • Positioning the graphs at the end of the results part complicated evaluation of results and comparison of information provided in the text and in graphs. I suggest to put graphs close to the text that describes its.
  • The providing of experiment pictures and/or schemes allows to increase the interestingness and digestibility of information for readers.
  • The provided results are interesting. However, in my humble opinion, authors do not describe exactly the design of biosensor, but rather provided an idea and validated it with good experimental results. In case of biosensor fabrication, I would like to see investigation performed with different nitrate concentrations or some similar experiments. This fact could be added to the title of article. For instance, the title could be “Concept of fluorescent transport activity biosensor for characterization of Arabidopsis3 activity of nitrate”.
  • The conclusion part should be added to the article.

Author Response

Response to Reviewer3:

Reviewer 3

Comments to the authors:

This paper reported the fabrication of fluorescent transport activity biosensor for characterization of Arabidopsis NPF1.3 activity of nitrate. Measurements of biosensor selectivity and additional experiments of nitrate transportation were performed. Overall, the work was well conducted, and the results were adequately presented and discussed. The manuscript can be considered for publication after minor corrections that should be revised by the authors.

Specific comments:

  • Positioning the graphs at the end of the results part complicated evaluation of results and comparison of information provided in the text and in graphs. I suggest to put graphs close to the text that describes its.

Thanks for the suggestions. We put all figures close to their related texts.

  • The providing of experiment pictures and/or schemes allows to increase the interestingness and digestibility of information for readers.’

Thanks for the suggestions. To make readers easier to follow and to increase their interestingness, the diagram of the workflow chart of sensor engineering and the experimental process was added as in Fig 1.

  • The provided results are interesting. However, in my humble opinion, authors do not describe exactly the design of biosensor, but rather provided an idea and validated it with good experimental results. In case of biosensor fabrication, I would like to see investigation performed with different nitrate concentrations or some similar experiments. This fact could be added to the title of article. For instance, the title could be “Concept of fluorescent transport activity biosensor for characterization of Arabidopsis3 activity of nitrate”.

Thanks of the suggestions. The concept to generate NFPs sensor comes from the NiTrac1. The sentences described the design of sensor as in line 146 “To capture substrate-dependent conformational rearrangements of transporter, NiTrac1 and PepTracs have been generated by sandwiching CHL1/NRT1;1/NPF6.3 and PTRs between a yellow acceptor and cyan donor fluorophore [15]. Regarding to our previous successful cases of NiTrac1 and PepTracs [15], it is likely that the other members of NPF family undergo conformational rearrangements during its transport cycle….The workflow diagram of engineering a biosensor as shown in Figure 1” were added.

As suggested, different concentrations of nitrate (2.5, 5, 10 mM of nitrate) were tested with NiTrac-NPF1.3 expressed yeast cells. The results showed that NiTrac-NPF1.3 can also response to different concentrations of nitrate. These results of NiTrac-NPF1.3 response to different concentrations of nitrate were added as in Fig 3.

In addition, the title is changed as suggested as “Concept of fluorescent transport activity biosensor for characterization of Arabidopsis NPF1.3 activity of nitrate”.

  • The conclusion part should be added to the article.

The conclusion was added as in line 298 “5. Conclusion. To sum up, by extension the idea of engineer transporter activity biosensor as NiTrac1 [15] to the member of NPF family, we developed a reporter, NiTrac1.3 biosensor, to report the activity of nitrate transporter NPF1.3 in vivo. As for next steps, we will deploy both NiTrac1 and NiTrac1.3 sensors in Arabidopsis roots to measure the nitrate fluxes in vivo and to study their regulation as well as to identify their regulators in vivo. Such development of activity biosensor and in vivo flux analyses will help us understanding the action of transporters and may also use as tools for drug development and discovery.” in the text.”
